# Alumina Ceramics for Armor Protection via 3D Printing Using Different Monomers

**DOI:** 10.3390/ma17112506

**Published:** 2024-05-23

**Authors:** Dongjiang Zhang, Zhengang Liang, Xin Chen, Chunxu Pang, Xuncheng Guo, Xiqing Xu

**Affiliations:** 1School of Equipment Engineering, Shenyang Ligong University, Shenyang 110159, China; 2Xi’an Modern Control Technology Research Institute, Xi’an 710065, China; 3School of Materials Science and Engineering, Chang’an University, Xi’an 710061, China

**Keywords:** alumina ceramics, 3D printing, monomer, interlayer gap, dynamic mechanical property

## Abstract

Alumina ceramic is an ideal candidate for armor protection, but it is limited by the difficult molding or machining process. Three-dimensional printing imparts a superior geometric flexibility and shows good potential in the preparation of ceramics for armor protection. In this work, alumina ceramics were manufactured via 3D printing, and the effects of different monomers on the photosensitive slurry and sintered ceramics were investigated. The photosensitive slurries using dipropylene glycol diacrylate (DPGDA) as a monomer displayed the optimal curing performance, with a low viscosity, small volume shrinkage and low critical exposure energy, and each of the above properties was conducive to a good curing performance in 3D printing, making it a suitable formula for 3D-printed ceramic materials. In the 3D-printed ceramics with DPGDA as a monomer, a dense and uniform microstructure was exhibited after sintering. In comparison, the sample with trimethylolpropane triacrylate (TMPTA) showed an anisotropic microstructure with interlayer gaps and a porosity of about 9.8%. Attributed to the dense uniform microstructure, the sample with DPGDA exhibited superior properties, including a relative density of 97.5 ± 0.5%, a Vickers hardness of 19.4 ± 0.8 GPa, a fracture toughness of 2.6 ± 0.27 MPa·m^1/2^, a bending strength of 690 ± 54 MPa, and a dynamic strength of 3.7 ± 0.6 GPa at a strain rate of 1200 s^−1^.

## 1. Introduction

Alumina ceramic is one of the most important ceramics in the fields of aerospace, machinery, metallurgy, semiconductors and armor protection [1,2,3,4], attributed to its advantages in hardness, wear resistance, high-temperature stability, and chemical corrosion resistance [5,6,7]. Serving as armor protective materials, alumina ceramics with a light weight counteract high-speed ammunition by virtue of their high hardness and high strength [8,9]. However, special shapes are generally required for armor protection systems, and armor protection ceramics with a high strength and hardness must be processed using diamond grinding tools [10,11], and the expensive price of machining processing reaches about 80% of the total cost [12]. In addition, there are some complex shapes in armor protection systems which cannot be manufactured using available manufacturing technologies, and new manufacturing technology is desired.

Three-dimensional printing [13,14,15], i.e., additive manufacturing technology, has a superior geometric flexibility, a high precision, and a high efficiency, making it a suitable molding method for ceramic materials with complex shapes. By virtue of 3D printing technology, ceramic materials have achieved a huge leap in geometric flexibility, and the production of ceramics with special shapes has become easier, which highly promotes the development of complex-shaped ceramic armor materials [16,17]. There are various kinds of 3D printing technologies for ceramic materials, including ink-jet printing, fused deposition modeling, selective laser sintering, digital light processing, and stereo lithography [18,19,20]. Compared to other available 3D printing technologies, digital light processing (DLP) 3D printing [21,22,23] has a high printing accuracy (within 10 μm) and a fast forming speed (up to 100 mm/h), making it popular for the preparation of complex-shaped ceramic parts.

DLP 3D printing relies on the curing reaction of the photosensitive resin into polymers under the irradiation of UV light, after which the ceramic powder is bonded into a green body with a certain strength and a smooth surface. Photosensitive resins [24,25] are mainly composed of oligomers, monomers, photoinitiators, dispersants, and other additives. Monomers undergo polymerization reactions due to the excitation of photoinitiators under UV irradiation, and the type of monomer directly affects the 3D printing performance of photosensitive resins, as well as the structure and properties of the 3D-printed ceramic materials. According to the Krieger–Dougherty model [26], the viscosity of ceramic slurries is dependent on the viscosity of photosensitive resins. Therefore, monomers with a low viscosity should be selected to obtain ceramic slurries with a high solid content and low viscosity to meet the requirements for 3D printing [27]. The number of active functional groups in monomers [28] plays an important role in photopolymerization reactions. Compared to monomers with a single-function group, those with multi-function groups significantly increase the density of crosslinks and improve the strength and hardness of printed green bodies. However, the increased number of functional groups also leads to an increased viscosity of the slurry, which is not conducive to 3D printing. Refractive index matching [29,30] is another factor in the selection of monomers. According to the Beer–Lambert law, the curing depth is inversely proportional to the square of the refractive index difference between a photosensitive resin and ceramic particles. Therefore, when the refractive index of monomers approaches that of ceramic particles, it can effectively limit scattering effects and increase the curing depth, thereby improving the printing success rate. In addition, the curing performance of monomers and the thermal decomposition of the cured products are also important factors to facilitate the subsequent degreasing process [31]. Overall, the selection of monomers directly affects the printing performance of photosensitive resins, as well as the properties and surface quality of the cured resin blocks, which further have a key impact on the structure and properties of 3D-printed ceramic materials.

To achieve complex-shaped alumina ceramics in armor protection systems, which cannot be manufactured using available manufacturing technologies, alumina ceramics were manufactured by DLP 3D printing in this work. Photosensitive resins with different monomers, i.e., dipropylene glycol diacrylate (DPGDA), 16-hexanediol diacrylate (HDDA), and trimethylolpropane triacrylate (TMPTA), were prepared. The influence of the monomer type on the 3D printing performance of the slurry was studied to explore the optimal formula of the photosensitive resin. The microstructures and properties of 3D-printed alumina ceramics were investigated.

## 2. Materials and Methods

### 2.1. Photosensitive Slurry

In the photosensitive resin, the oligomer employed in this work was polyurethane acrylate (PUA), the photoinitiator was TPO-L, and the effects of monomers on the photosensitive resin slurry were investigated using DPGDA, HDDA, and TMPTA as monomers, respectively. The oligomer and monomer were mixed in mass ratio of 1:1, and then 4 wt.% of photoinitiator (TPO-L) was added. The photosensitive resin premix was obtained after the mixture was stirred for 2 h and then was left to rest for 24 h until the bubbles disappeared completely. The above organic materials were all analytically pure and purchased from Shanghai Yinchang New Materials Co., Ltd., Shanghai, China.

Alumina powders were added into the photosensitive resin premixture with a solid loading of 56 vol.%, and stirred to uniformity via ball milling for 180 min. Alumina powders with a purity of >99.99% were purchased from Sumitomo Co., Ltd., Osaka, Japan, and the particle size distribution, measured using a laser particle size analyzer (Masrer Sizer 2000, Malvern Panalytical, Malvin, UK), in Figure 1, showed a mean diameter of 4.0 μm. The photosensitive ceramic slurry was finally obtained after filtering and vacuum defoaming.

### 2.2. Three-Dimensional Printing and Sintering

The mixed ceramic slurry was poured into a DLP 3D printer (Autocera-M, Beijing Shiwei Technology, Beijing, China), and layer by layer, 3D printing was performed based on a pre-established digital model, as expressed by the diagrammatic sketch in Figure 2. After printing, the green body was washed in an ultrasonic cleaner to remove the excess slurry and dried in a drying oven at 120 °C in air. The dried body was then placed in a muffle furnace for degreasing and sintering under an air atmosphere, during which the final sintering temperature was 1700 °C, which was held for 4 h; the detailed sintering program is displayed in Section 3.2.

### 2.3. Characterization and Testing of the Slurry and Ceramics

The viscosity of the photosensitive resin and ceramic slurry was measured using a constant-temperature viscometer (THS-NDJ-5S, Shenzhen, China), with a rotor speed of 50 r/min at 25 °C. The dimensions of 3D-printed green body were measured using Vernier calipers with an accuracy of 0.02 mm, and the volume shrinkage after photopolymerization and sintering was determined in comparison to the digital model of 55 mm × 10 mm × 4 mm.

The thickness of the curing layer was evaluated using a spiral micrometer after exposure and removal of uncured photosensitive resin. The measurement was carried out taking the average value at the center point and the four corner positions of the square cured film.

The critical exposure and critical transmission depth of the photosensitive resins with different monomers were determined based on the Beer–Lambert theorem [32,33]:(1)$$C_{d}=D_{p}ln(E_{i}/E_{c}),$$
where *C_d_* is the curing depth by photopolymerization, i.e., the thickness measured by curing a photosensitive resin at a certain exposure energy density. *D_p_* is the critical transmission depth, i.e., the depth of light penetration that triggers the critical exposure energy required for photopolymerization. *E_c_* is the critical exposure energy, below which the exposure energy is not able to cure the photosensitive resin. *E_i_* is the energy of the input light source on the resin surface, which depends on the optical machine power *W*_0_ and the exposure time *t*:(2)$$E_{i}=W_{0}\times t,$$

The curing thickness of the resin slurry was measured under different input energies and fitted versus lnE_i_; the values of *D_p_* and *E_c_* can be determined according to the slope and intercept values.

The thermogravimetric curve (TG) and differential thermogravimetric curve (DTG) of the 3D-printed green body were obtained using a thermal analyzer (TG209F1, NETZSCH, Selb, Germany) with heating rate of 5 °C/min under air atmosphere.

Scanning electron microscopy (SEM, JSM-5600LV, JEOL, Tokyo, Japan) was performed to analyze the microstructure of the 3D-printed samples before and after sintering. The grain sizes of sintered ceramics were measured from images using the linear intercept method [34].
(3)$$\bar{D}=1.56\frac{C}{MN}$$
where $\bar{D}$ is the average grain size, C represents the measuring line length in the micrograph, M is the magnification times, and N is the number of the sections in the micrograph.

Archimedes’ method was carried out to obtain the bulk density and apparent porosity of the 3D-printed ceramics samples after sintering according to ASTM C373 [35].

Vickers indentation tests were carried out on the polished surfaces of sintered alumina samples using a microhardness machine (HXD-1000TM, Shanghai Changfang Optical Instrument Co. Ltd., Shanghai, China) to study the Vickers hardness (H_V_) of the samples, during which 49 N was loaded and held for 15 s.

Based on the diagonal length of indentation, H_V_ was evaluated through the equation:(4)$$H_{V}=1.854\frac{P}{d^{2}}$$
where P represents the load and d expresses the mean value of the diagonal length. Each value of H_V_ was taken from the average value of 20 indentations.

Three-point bending was carried out to measure the flexure strength (σ) of the samples using an electromechanical testing machine (Instron5500R, Norwood, MA, USA). The sintered bars had dimensions of 40 mm × 3 mm × 2 mm and a supporting span of 15 mm, and the loading rate was 1 mm/min. The flexure strength (σ) was determined based on the equation:(5)$$\sigma =\frac{3PL}{2bd^{2}}.$$
where *P* is the load at which the bars broke down, *L* is the support span, *b* is the specimen width, and *d* is the specimen thickness. Each measurement was performed on eight specimens, and the mean values of these measurements were taken as the three-point bending values.

The fracture toughness was also measured through single-edge notched beam specimen techniques. The dimensions of the testing bars were 30 × 6 × 4 mm in length, thickness and width, with a notch depth of 2.5 mm, and the supporting span in three-point bending was 20 mm. The fracture toughness through the SENB method could be expressed by:(6)$$K_{IC}=Y\frac{3PL}{2bh^{2}}\sqrt{a}.$$
(7)$$Y=1.99-2.47\frac{a}{h}+12.97{\left(\frac{a}{h}\right)}^{2}-23.17{\left(\frac{a}{h}\right)}^{3}+24.80{\left(\frac{a}{h}\right)}^{4}$$
where P is the largest load before fracture, L is the supporting span, *b* and *h* are the width and thickness of the tested bars, and a is the notch depth. Each value was determined by taking the average of five specimens.

A split Hopkinson pressure bar (SHPB) test was performed on the alumina ceramics after sintering to reveal their dynamic mechanical properties. To ensure that the test method was suitable for brittle materials, a copper pulse shaper was located between the striker and the incident bars. To prevent the ceramic samples from indenting the incident and transmission bar faces, tungsten carbide platens were placed between the samples and the bars’ faces. These platens had the same impedance as the bars. The stress wave spreading along the SHPB was recorded, and the load and displacement of the pressure bar were plotted as a function of time, after which the dynamic stress–strain curves were obtained for the samples.

## 3. Results and Discussion

### 3.1. Properties of Resins and Slurries

The viscosities of photosensitive resins and ceramic slurries with different monomers are shown in Figure 3. As HDDA was employed as a monomer, the photosensitive resin without ceramic particles showed the lowest viscosity of 360 mPa·s, and the ceramic slurry with a solid loading of 56 vol.% showed the lowest viscosity of 1620 mPa·s; meanwhile, the photosensitive resin and ceramic slurry showed the largest viscosity values of up to 2910 mPa·s and 6540 mPa·s, respectively, with the monomer of TMPTA. It is noted that TMPTA is an active agent with three functional groups and has the highest molar mass; therefore, the photosensitive resin with the monomer of TMPTA showed a higher viscosity than the others. Both HDDA and DPGDA have two functional groups, and the molecular weight of DPGDA (242) is slightly larger than that of HDDA (224); therefore, the photosensitive resin and slurry with the monomer of HDDA showed the best dispersivity and lowest viscosity.

Figure 4 shows the effect of monomer type on the volume shrinkage during UV curing of photosensitive resins and ceramic slurries. It was observed that the volume shrinkage of photosensitive resin and ceramic slurries reached the maximum values of 4.36% and 1.43%, respectively, when TMPTA was employed as the monomer. Meanwhile, minimum values of 1.72% and 0.46% were recorded when DPGDA was used as the monomer. As is known, the monomer of TMPTA with three functional groups has higher numbers and densities of double bonds than HDDA and DPGDA with two functional groups. In the process of UV curing, larger quantities of crosslinking points were generated and more molecular bonds transform from van der Waals bonds into covalent bonds, resulting in a larger volume shrinkage [28]. Although both HDDA and DPGDA have two functional groups, HDDA has a smaller molecular weight, which led to a better fluidity during UV curing and resulted in a higher volume shrinkage.

The curing depth of the photosensitive resins as a function of different monomers was measured under certain exposure energy densities, as listed in Table 1. For each monomer, regardless of whether there were two or three functional groups, the curing depth showed a monotonic increase with an increasing exposure energy density. When the exposure energy density was 150 mJ/cm^2^, the values of curing thickness were 0.56 mm, 0.68 mm and 0.85 mm for photosensitive resins with monomers of HDDA, DPGDA and TMPTA, respectively.

The curing depth (*Cd*) was plotted versus the natural logarithm of the exposure energy (*lnE*), and linear fitting was performed and displayed in Figure 5a. The data exhibit a good linear fit, with an R^2^ larger than 0.97, and the slight deviations at a low exposure energy are attributed to the errors in measurement of the curing depth. It is seen that the slope of the fitting line of the TMPTA photosensitive resin system was much larger than those of the other two monomers, and the fitting lines of HDDA and DPGDA photosensitive resins were nearly parallel, suggesting a similar critical transmission depth. The critical transmission depth and critical exposure energy were determined based on the slope and intercept of the fitting line, as plotted in Figure 5b. It was observed that the critical exposure energies of photosensitive resins with monomers of DPGDA, HDDA, and TMPTA were 17.7, 22.8, and 27.5 mJ/cm^2^, respectively, and the critical transmission depths were 0.31, 0.30, and 0.48 mm. Generally speaking, a small critical exposure energy means an easy transformation from a photosensitive resin to the bulk under UV curing [36]. Therefore, the monomer of TMPTA with three functional groups made the curing of photosensitive resin more difficult, while DPGDA made it easier.

### 3.2. Sintering Procedure

TG and DTG curves of the 3D-printed green body with monomers of DPGDA are plotted in Figure 6a. It is shown that the green body exhibited a significant mass loss below 600 °C, attributed to the combustion of organic matter in the photosensitive resin. Above 600 °C, the TG curve remains unchanged, meaning complete thermal decomposition of the organic resins, and the mass loss was about 23 wt.% according to the TG curve. Based on the DTG curve, the degreasing speed showed maximum values at 262 °C, 365 °C, and 505 °C, related to the combustion of the oligomer (PUA), monomer (DPGDA) and photoinitiator (TPO-L). During degreasing, appropriate holding for 1 h at 262 °C, 365 °C, and 505 °C is required to ensure full elimination of the organic resin without damaging the ceramic bodies. After the degreasing process was completed, the sample underwent a quick heat treatment at 1200 °C along with a short holding of 1 h; after which, the sample was further heated to 1700 °C and held for 4 h, as shown in the detailed sintering procedure in Figure 6b. Similar TG and DTG tests were performed on the green body with monomers of HDDA and TMPTA, Figure 6c,d, and the related sintering procedures were determined.

### 3.3. Microstructure of the Ceramics

Figure 7 shows the SEM images recorded from the natural surface of the green bodies 3D printed with different monomers, showing the surface was neither fractured nor polished. When DPGDA is employed as a monomer, the 3D-printed green body showed uniform microstructures with tiny interlayer cracks (marked by red arrows in Figure 7a). The sample using HDDA as a monomer showed an anisotropic microstructure with thin parallel cracks (marked by red arrows) between the printing layers (Figure 7b). In the sample with monomers of TMPTA, the microstructure was rougher, and the interlayer cracks developed into large gaps with a width of up to 40 μm (Figure 7c); similar interlayer gaps have been reported in various studies [37,38,39]. Generally speaking, the printing accuracy indicates the resolution of UV and the responses to the formation error when the photosensitive resin is fully cured. The interlayer gaps were attributed to incomplete curing, which was not dependent on the accuracy of DLP 3D printing. The cracks or gaps were attributed to incomplete curing of the photosensitive resin, a part of which remained uncured and turned into interlayer gaps after cleaning. According to Figure 5, the curing of TMPTA is difficult; therefore, the green body from photosensitive resins with TMPTA contained wide gaps between the printed layers.

Figure 8 shows the SEM images of the natural surface of the sintered samples 3D printed with different monomers. After sintering, the interlayer cracks disappeared completely when DPGDA and HDDA were employed as monomers, Figure 8a,b. Generally speaking, multiple sintering mechanisms are involved in the solid-state sintering of alumina ceramics, including evaporation–condensation, diffusion mass transfer, viscous and plastic flow, and dissolving–precipitation [17]. It is confirmed that the particles re-arranged and solid-state diffusion between different printing layers took place during sintering, which removed the directionally arranged pores between the printing layers. Therefore, the microstructures were dense and uniform. In the sample with TMPTA (Figure 8c), the grain size was slightly larger and the porosity was significantly higher than that in Figure 8a,b. The gaps attributed to the oriented arrangement of residual pores between the printing layers are still existent, even though significant narrowing via solid-state diffusion occurred during sintering. It is suggested that the large interlayer gaps are unable to be removed during sintering, as the width was up to 40 μm in the green body.

Detailed properties of the 3D-printed ceramics with different monomers after sintering were measured and are listed in Table 2. Based on the Archimedes method, the relative densities were 97.5 ± 0.5% and 96.7 ± 0.6% for the ceramics with monomers of DPGDA and HDDA, respectively, and these values were in agreement with the dense microstructures in the SEM images (Figure 8a,b). Due to significant densification, the volume shrinkages were up to 42.5 ± 0.8% and 42.0 ± 1.0%. As the monomer was TMPTA, the relative density was as low as 90.2%, and the volume shrinkage decreased to 37.8 ± 1.4%, which was attributed to the residual pores in Figure 8c. Regardless of the monomers, the samples showed a similar grain size of about 16 μm. As listed in Table 2, the sample with TMPTA showed poor mechanical properties due to the low relative density of 90.2% and significant interlayer gaps, as the pores acted as crack origins and the interlayer gaps provided an easy path for crack spreading. The alumina ceramics with monomers of DPGDA and HDDA showed optimal mechanical properties due to the dense uniform microstructures with similar relative densities and grain sizes. Even though no significant difference is observed between samples with DPGDA and HDDA, the mechanical properties of the sample with DPGDA are slightly better, with a Vickers hardness of 19.4 ± 0.8 GPa, a fracture toughness of 2.6 ± 0.27 MPa·m^1/2^, and a bending strength of 690 ± 54 MPa. Common commercial alumina ceramics for armor show a hardness of 12~20 GPa, a bending strength of 250~550 MPa, and a fracture toughness of 2.5~4.7 MPa·m^1/2^ [9], and the mechanical properties of sample with DPGDA were comparable to these commercial armor ceramics.

An SHPB test was conducted to investigate the dynamic mechanical properties of the alumina ceramics after sintering, and the strain rate in the SHPB test was 1200 s^−1^. Each compressive test was carried out on five samples for each composition, and the compressive stress can be determined by the mean value and standard scatter. However, the dynamic stress–strain curve is composed of continuity points, and it cannot be formed by average values. Therefore, we show a representative stress–strain curve for each test in Figure 9.

Regardless of the monomers, the alumina ceramics showed a near-linear deformation of about 3% before the maximum stress, which is regarded as elastic deformation and is the major mode of deformation without plastic deformation. After elastic deformation, nonlinear deformation with a strength reduction was observed, which was related to crack initiation and propagation through the samples. The stress–strain curves in this work displayed typical features of brittle alumina ceramics, as the nonlinear deformation after the maximum stress is unrelated to plastic deformation, even though the total strain was up to 6%. Compared to the static mechanical tests, the values of maximum stress and strain were very high in dynamic tests with high strain rates, similar to other studies [40,41]. Considering that the major mechanism of deformation in brittle materials, including ceramics, is the propagation of cracks [41,42], in dynamic mechanical tests, the speed of crack propagation was much lower than the loading rate, and the cracks are unable to propagate quickly. The delay of crack propagation led to delayed sample fracture, achieving higher strength and strain values for the alumina ceramics at higher strain rates [43,44].

For alumina ceramics with different monomers, the sample with DPGDA showed a superior performance with a dynamic strength of 3.7 ± 0.6 GPa due to the dense and uniform microstructure in Figure 8a, which was comparable to those in literatures [40,44]. In comparison, the sample with TMPTA showed the worst dynamic performance, with a dynamic strength of 2.5 ± 0.5 GPa, attributed to the inhomogeneous microstructure with interlayer gaps and a porosity of about 9.8% in Figure 8c. It was suggested that the pores acted as the crack origin and the interlayer gaps provided an easy path for crack spreading, which cooperatively deteriorated the dynamic mechanical properties of the alumina ceramics. It was noted that the stresses in Figure 9 were one order of magnitude higher compared to the dynamic stress and bending stress in Table 2. On the one hand, the quasi-static compressive strength of ceramics is much higher than the bending strength, as widely reported. For ceramics with a bending strength of about 400 MPa, the compressive strength can be up to 2 GPa [9]. On the other hand, the compressive strength is much higher in dynamic tests than in quasi-static state due to the delay of crack propagation at high strain rates [43,44].

## 4. Conclusions

In this work, alumina ceramics for armor protection were manufactured via 3D printing, and the effects of different monomers, i.e., DPGDA, HDDA, and TMPTA, on the performance of the photosensitive slurry and the microstructures and properties of the 3D-printed alumina ceramics were investigated. The conclusions are as follows:(1)The photosensitive slurries using DPGDA as a monomer with a solid loading of 56 vol.% displayed the optimal curing performance, with a low viscosity of 2910 mPa·s and a small volume shrinkage of 0.46% during 3D printing, and the photosensitive resin with monomers of DPGDA showed the lowest critical exposure energy of 27.5 mJ/cm^2^, suggesting easy curing to form the solid state. Each of the above properties is conducive to a good curing performance in 3D printing, making it a suitable formula for the 3D printing of ceramic materials.(2)In the 3D-printed sample using DPGDA as a monomer, microstructures with tiny interlayer cracks were exhibited in the green body, and the cracks disappeared completely after sintering due to particle re-arrangement and solid-state diffusion between different printing layers, resulting in a dense and uniform microstructure. In comparison, when TMPTA was employed as a monomer, the green body showed an anisotropic microstructure with serious interlayer gaps with widths of up to 40 μm, which was attributed to the difficulty in curing TMPTA with three functional groups. Furthermore, the sintering driving force was unable to remove the wide interlayer gaps, leading to an inhomogeneous microstructure with residual interlayer gaps and a porosity of 9.8%.(3)The different monomers had a further significant effect on the mechanical properties of alumina ceramics after sintering. The sample with monomers of TMPTA showed poor mechanical properties, attributed to the low relative density and serious interlayer gaps in the microstructure. In comparison, the dense uniform microstructure in the sample with DPGDA as a monomer contributed to its superior properties, with a relative density of 97.5 ± 0.5%, a Vickers hardness of 19.4 ± 0.8 GPa, a fracture toughness of 2.6 ± 0.27 MPa·m^1/2^, a bending strength of 690 ± 54 MPa, and a dynamic strength of 3.7 ± 0.6 GPa at a strain rate of 1200 s^−1^, comparable to the properties of commercial armor ceramics.

## Figures and Tables

**Figure 1 materials-17-02506-f001:**
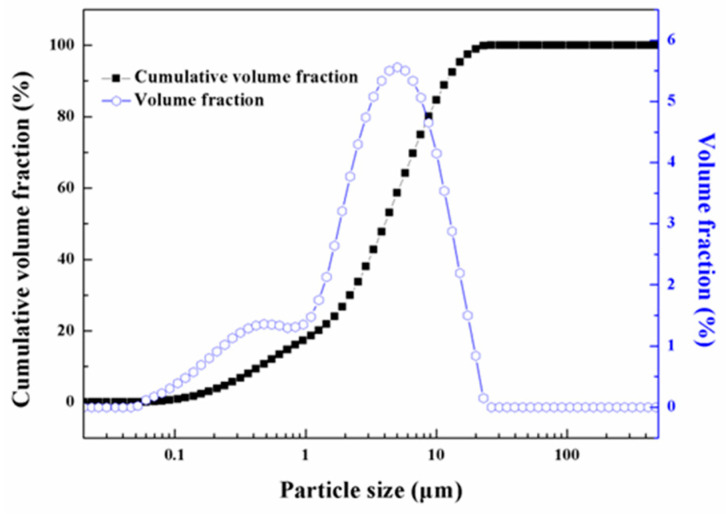
Particle size distribution of alumina powders.

**Figure 2 materials-17-02506-f002:**
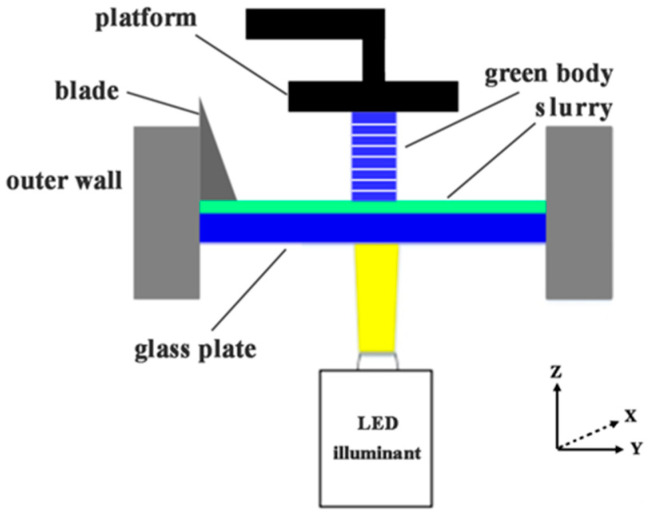
Schematic diagram of the DLP 3D printing process.

**Figure 3 materials-17-02506-f003:**
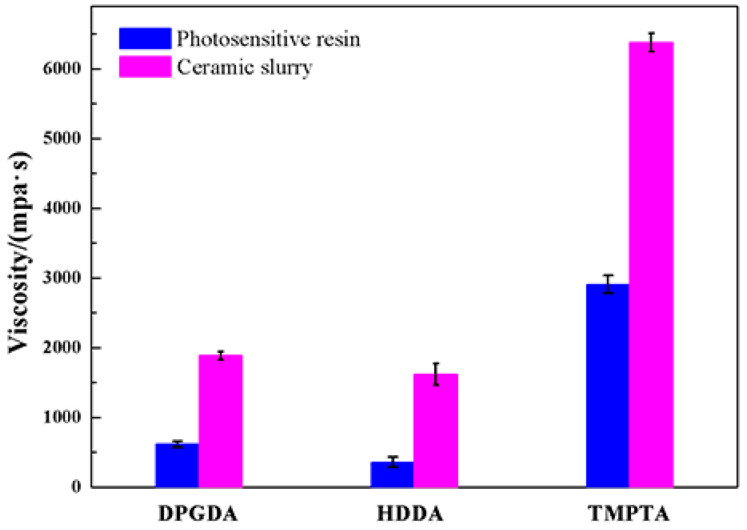
The viscosities of photosensitive resins and ceramic slurries with different monomers.

**Figure 4 materials-17-02506-f004:**
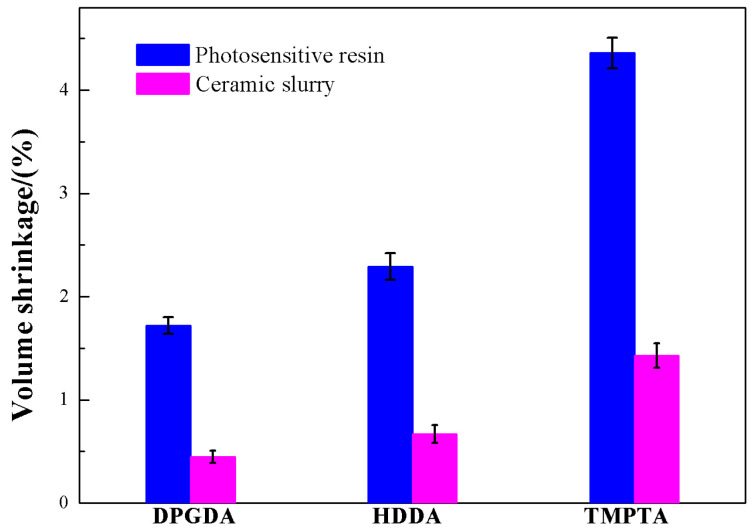
The volume shrinkages of photosensitive resins and ceramic slurries with different monomers during UV curing.

**Figure 5 materials-17-02506-f005:**
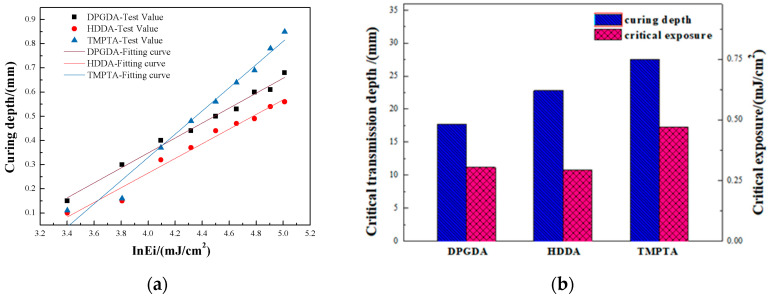
(**a**) The curing depth plotted versus *lnEi* along with the linear fitting; (**b**) the critical transmission depth and critical exposure energy of photosensitive resins with different monomers.

**Figure 6 materials-17-02506-f006:**
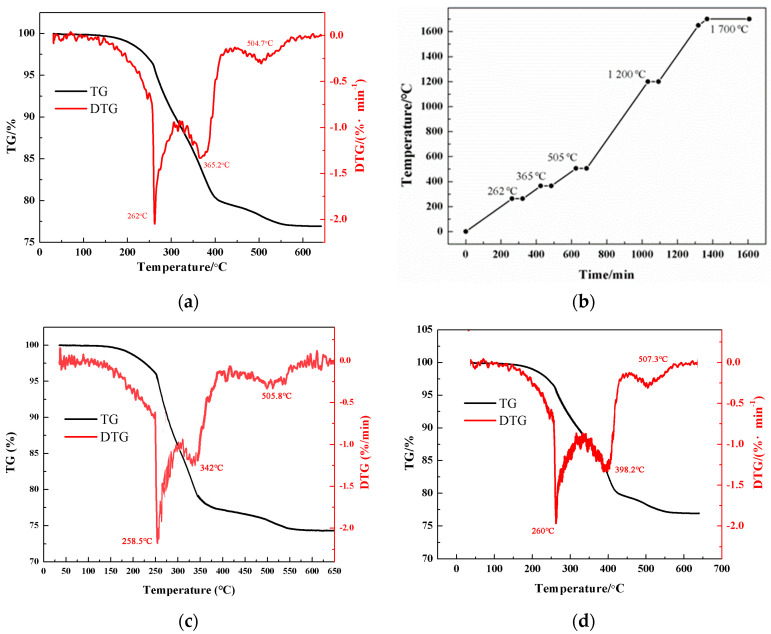
(**a**) TG and DTG curves of the 3D-printed green body with monomers of DPGDA; (**b**) the detailed sintering procedure of ceramics; (**c**) TG and DTG curves of the green body with HDDA; (**d**) TG and DTG curves of the green body with TMPTA.

**Figure 7 materials-17-02506-f007:**
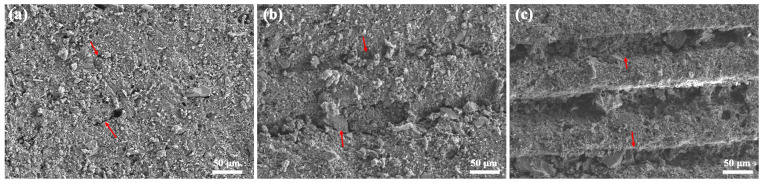
SEM images recorded from the natural surface of the green bodies 3D printed with different monomers: (**a**) DPGDA; (**b**) HDDA; (**c**) TMPTA.

**Figure 8 materials-17-02506-f008:**
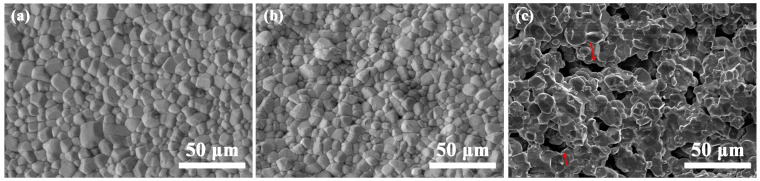
SEM images recorded from the natural surface of sintered samples 3D printed with different monomers: (**a**) DPGDA; (**b**) HDDA; (**c**) TMPTA.

**Figure 9 materials-17-02506-f009:**
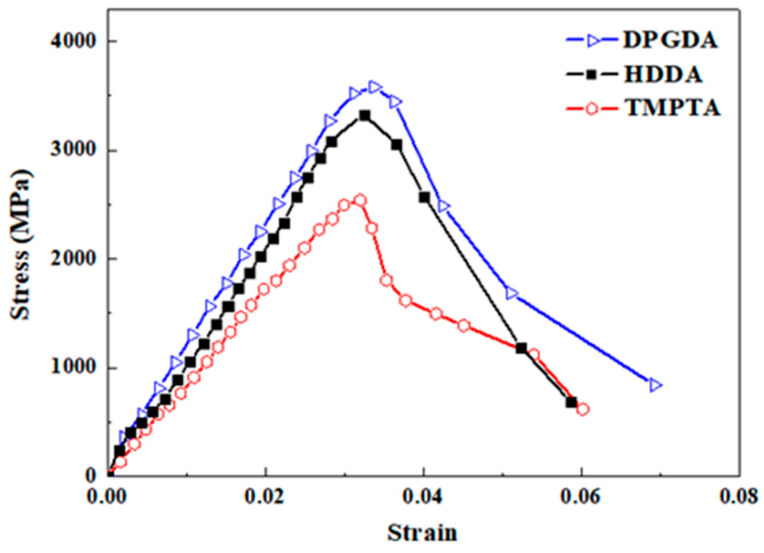
Dynamic stress–strain curves of the sintered alumina ceramics constructed by 3D printing with different monomers: DPGDA; HDDA; TMPTA.

**Table 1 materials-17-02506-t001:** Curing depth (mm) of photosensitive resins with different monomers under a certain exposure energy (mJ/cm^2^).

		Ei	30	45	60	75	90	105	120	135	150
	Cp										
Monomer											
DPGDA			0.15	0.3	0.4	0.44	0.5	0.53	0.6	0.61	0.68
HDDA			0.1	0.15	0.31	0.37	0.44	0.47	0.49	0.54	0.56
TMPTA			0.11	0.16	0.37	0.48	0.56	0.69	0.69	0.78	0.85

**Table 2 materials-17-02506-t002:** Properties of alumina ceramics with different monomers.

Monomer	Relative Density (%)	Volume Shrink Age (%)	Grain Size (μm)	HV (GPa)	KIC (MPa·m^1/2^)	σ (MPa)
DPGDA	97.5 ± 0.5	42.5 ± 0.8	15.9 ± 2.2	19.4 ± 0.8	2.6 ± 0.27	690 ± 54
HDDA	96.7 ± 0.6	42.0 ± 1.0	15.7 ± 4.7	18.8 ± 1.1	2.4 ± 0.35	660 ± 76
TMPTA	90.2 ± 1.3	37.8 ± 1.4	16.9 ± 6.4	15.5 ± 2.1	1.9 ± 0.52	510 ± 76

## Data Availability

Data are contained within the article.

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
