# Peer review of "Alumina Ceramics for Armor Protection via 3D Printing Using Different Monomers"

_materials, 2024, doi:10.3390/ma17112506_

Round 1

Reviewer 1 Report

Comments and Suggestions for Authors

My comments and suggestions are highlighted in the attached file.

It appears that the goal is unclear. 3D printing is used for complex structures but very simple shapes were made. The authors seem to not be acquainted with the sintering mechanisms of alumina ceramics. The discussion of data related to the dynamic mechanical testing reveals lack of in-depth knowledge of fracture mechanics of brittle materials. These aspects should be carefully revised.  

Comments on the Quality of English Language

fair quality. English is not my mother tongue.

Author Response

Dear Editor and Reviewers,

We have tried our best to revise and improve the manuscript and made great changes in the manuscript according to your good comments. This file mainly explains how we have addressed the reviewer's concerns and the revised manuscript are marked at places where the content has been revised. We appreciate for your work earnestly, and hope that the corrections will meet with your approval.  

Yours Sincerely

Xiqing Xu

-------------------------------------------------------------------------------------

The following is a point-to-point response to the comments.  

Reviewer 1:

1. It appears that the goal is unclear.

Reply:

Thank you very much for your constructive suggestion. The goal is “To avoid machining processing in preparation of complex-shaped alumina ceramics for armor protection”. We have added the goal in the revised manuscript in Page 2.

2. 3D printing is used for complex structures but very simple shapes were made.

Reply:

Thank you for your suggestion. Simple shapes with standard sizes are necessary in  testing of mechnical properties. Therefore, simple shapes were made in this work. In fact, in most literatures about 3D printing, the samples are in simple shapes to meet the test demand.

3. The authors seem to not be acquainted with the sintering mechanisms of alumina ceramics.

Reply:

Thank you very much for your constructive suggestion. The sintering mechanisms of alumina ceramics were provided in the new manuscript based on literature. (highlighted in yellow in Page 8)

4. The discussion of data related to the dynamic mechanical testing reveals lack of in-depth knowledge of fracture mechanics of brittle materials.

Reply:

Thank you very much for your constructive suggestion. Detailed discussion on dynamic mechanical testing was revised in the manuscript. (marked in red in page 9)

5. My comments and suggestions are highlighted in the attached file.

Reply:

Thank you very much for your careful guidance, we have made point-to-point responses in the following PDF file, please find it in the attachment.

Reviewer 2 Report

Comments and Suggestions for Authors

1) Regarding the English language, minor editing of the English language is required: On page 1, row 11 the word ceramic is doubled, on the same page, row 40 the word making is doubled.

2)Including the best-achieved value for relative density in the abstract is recommended. 

3) Could authors provide some connections between results obtained for viscosity and curing thickness, and volume shrinkage and curing as well between curing thickness and intelayer gaps dishcharge ?

4) Discusions should be provided by authors between results shown in table 1. and mechanical properties.

Overall, papper gives nice and clear presentation effects of different monomers regarding their structure (ie funct. groups) and curing process which reflects further on relstive density and mechanical properties. 

Comments on the Quality of English Language

Regarding English language, some minor corrections should be made:

On the page 1, row 11, word ceramic is doubled, on the same page in the row 40, word making is also doubled.

No other issues regarding English language.

Author Response

Please find the point-to-point response in the attachment.

Round 2

Reviewer 1 Report

Comments and Suggestions for Authors

The revised version is substantially better than the original one. I do appreciate the effort of the author's to improve the quality of the manuscript. However, some issues need still to be addressed (please see attached file).

I have found several reviews papers on this topic of research, which are not mentioned in the references, e.g. https://doi.org/10.1016/j.ceramint.2021.06.066

The scope of the work remains unclear and the reasons highlighted are not the most appropriate ones (see below).

Some technical details must be provived, particularly concerning fracture toughness measurements.

Data on TG/DSC requires further analysis: three different temperatures are related to which reactions?

Injection molding of complex alumina parts is quite common and therefore the reason for performing the current study cannot be related to avoid machining processing. I suggest that you consider complex shapes which cannot be manufactured using available manufacturing technologies.

How does the technique applied compares to other available 3D-printing technologies. Probably the answer to my previous comment lies in the advantages or drawbacks of the current technique compared to others.

The title is unclear. Please consider revising it. What does it mean "designed slurry"?

The volume shrinkage presented is of the printed parts. Densification will result in reduction in size larger than these. It is not reported and needs to be taken into account when designing complex parts.

The green body exhibited significant mass loss at 262 ℃, 365 ℃, and 505 ℃. Not only the composition of the curve shown is not provided, but also no explanation for the degradation of the resin is given.

Degreasing, often called defatting or fat trimming, is removal of fatty acids from an object. In this particular case, this is not the case. Please specify exactly what happens when the 3D parts are heated under air (i.e. DSC data needs to be discussed). The same trend occurs for all the resins used? 

What does it mean natural surface of the green bodies (line 244)?

In Fig. 8, one cannot see massive pores between layers. Please consider revising it.

Data in Table 2 seems not consistent with curves shown in Fig. 9. Wonder why? Stresses are one order of magnitude higher when comparing dynamic stress versus bending stress. This is probably related to the fact that the split Hopkinson pressure bar (SHPB) has been in used to measure the compression response, mainly of metals. Which is the scatter of the data shown in Fig. 9? Are results obtained comparable to those reported in the literature? Ceramic do not undergo plastic deformation. Is this test method suitable for such brittle materials? Did you apply the modifications required for testing brittle ceramics?

(e.g. see https://doi.org/10.31399/asm.hb.v08.a0003299)

If printing accuracy is about 10 microns, the conclusion that the gap of interlayers of 40 microns is due to difficult curing must be clarified.

Finally, Vickers hardness refers to macrohardness measurements carried out with forces ranging from 5 to 100 kgf. In this study, HV1 is considered as low-force hardness test. In reality, Vickers data ought to be compared for measurements performed using the same load. The smaller the indentation, the bigger the error associated to hardness measurement. HV does not vary with load, so the fact that a lower hardness recorded for materials made for PMPTA is certainly related to the remaining porosity.

Comments on the Quality of English Language

Some mispellings are highlighted in the attached file. Please check the English language; e.g. leaded does exist, but it refers to a poison related to the element lead.

Author Response

Dear Editor and Reviewers,

We have tried our best to revise and improve the manuscript and made great changes in the manuscript according to your good comments. We appreciate for your work earnestly, and hope that the corrections will meet with your approval. Please find the point-to-point response to the comments in the attachment.

Yours Sincerely

Xiqing Xu
